# Upregulation of PODXL and ITGB1 in pancreatic cancer tissues preoperatively obtained by EUS-FNAB correlates with unfavorable prognosis of postoperative pancreatic cancer patients

Keisuke Taniuchi[1]*, Makoto Ueno[2], Tomoyuki Yokose[3], Masahiko Sakaguchi[4], Reiko Yoshioka[1], Mitsunari Ogasawara[1], Takuhiro Kosaki[5], Seiji Naganuma[6], Mutsuo Furihata[6]

1 Department of Gastroenterology and Hepatology, Kochi Medical School, Kochi University, Nankoku, Kochi, Japan, 2 Department of Gastroenterology, Hepatobiliary and Pancreatic Medical Oncology Division, Kanagawa Cancer Center, Yokohama, Kanagawa, Japan, 3 Department of Pathology, Kanagawa Cancer Center, Yokohama, Kanagawa, Japan, 4 Faculty of Information and Communication Engineering, Osaka Electro-Communication University, Osaka, Japan, 5 Department of Endoscopic Diagnostics and Therapeutics, Kochi Medical School, Kochi University, Nankoku, Kochi, Japan, 6 Department of Pathology, Kochi Medical School, Kochi University, Nankoku, Kochi, Japan

* ktaniuchi@kochi-u.ac.jp

**Data Availability Statement:** All relevant data are within the paper.

## Abstract

The upregulation of PODXL and ITGB1 in surgically resected pancreatic cancer tissues is correlated with an unfavorable postoperative prognosis. The aim of this study was to investigate whether PODXL and ITGB1 are useful preoperative markers for the prognosis of postoperative pancreatic cancer patients in comparison with the TNM staging system. Immunohistochemistry was performed using anti-PODXL and anti-ITGB1 antibodies on 24 pancreatic cancer tissue samples preoperatively obtained by endoscopic ultrasound-guided fine-needle aspiration biopsy. Cox proportional hazards regression analysis was performed to investigate if the UICC TNM stage and upregulation of PODXL and ITGB1 were correlated with postoperative overall survival rates. Univariate analysis revealed that PODXL, TNM stage, lymphatic invasion and the combination of PODXL with ITGB1 are correlated with postoperative survival. Multivariate analysis demonstrated TNM stage and the combination of PODXL with ITGB1 to be correlated with postoperative survival, and the combination of PODXL with ITGB1 most accurately predicted the postoperative outcomes of pancreatic cancer patients before resection. Therefore, upregulation of PODXL and ITGB1 may indicate preoperative neoadjuvant therapy for pancreatic cancer patients by accurately predicting the postoperative prognosis.

## Introduction

Although there have been recent advances in novel cancer therapies, pancreatic ductal adenocarcinoma (PDAC) has the lowest 5-year survival rate [1, 2]. Only 15–20% of PDAC patients

**Funding:** This study was supported by Grants-in-Aid for Scientific Research (KAKENHI): 17K09463 (kt), 19K07461 (mf), 20K07806 (ry), 20K08359 (kt) and 20K07699 (mo), and AMED under Grant Number JP19lm0203007 (kt). The funders had no role in study design, data collection and analysis, decision to publish, or preparation of the manuscript.

**Competing interests:** The authors have declared that no competing interests exist.

are diagnosed with potentially resectable disease, approximately 35% with localized unresectable disease, and approximately 50% with end-stage disease [3, 4]. Surgery is the only potentially curative treatment for PDAC [3]. However, the prognosis of resected PDAC patients remains poor due to the high rate of local recurrence and/or distant metastases, and the 5-year survival rate for these patients is only 20% [5]. The resection margin is the most important factor related to the prognostic outcome of resected PDAC patients, and a positive resection margin usually results in a higher risk of local recurrence and distant metastasis [6], but adjuvant chemotherapy after surgical resection can improve overall survival [7]. Borderline resectable or locally advanced PDAC is radiographically defined with or without vascular involvement, but the definitions for what constitutes borderline resectable and locally advanced PDAC have historically varied across institutions [8]. Recently, neoadjuvant therapy prior to surgical resection for PDAC patients with borderline resectable and locally advanced disease was reported to achieve tumor downstaging with the aim of secondary curative intent surgery [9]. Palliative chemotherapy and best supportive care remain the only options for metastatic PDAC patients [10].

Immunohistochemical analysis of surgical specimens from 102 PDAC patients revealed the upregulation of podocalyxin-like protein (PODXL) in 70.6% of the surgical specimens [11]. PODXL expression significantly correlates with histological grade, but it is not associated with other clinicopathological features such as clinical stage, venous invasion and lymphatic invasion [11]. Similar effects occurred with integrin β1 (ITGB1), as ITGB1 is upregulated in 67.6% of the surgical specimens of PDAC, and ITGB1 is not associated with any clinicopathological features [12]. Importantly, the combination of PODXL with ITGB1 immunohistochemically scored using resected PDAC tissues accurately predicts the postoperative prognosis of PDAC patients better than the Union for International Cancer Control (UICC) tumor node metastasis (TNM) staging system [12]. TNM staging was reported as an independent prognostic factor affecting the prognosis of patients with PDAC [13]. In addition, 86 of 102 PDAC patients received adjuvant chemotherapy, which does not correlate with the postoperative prognosis [12]. PODXL functions in promoting the invasion of PDAC cells by binding to the cytoskeletal protein gelsolin [11]. ITGB1 was reported to promote cell invasion and tumor metastasis of PDAC cells [14], thus ITGB1 is a therapeutic target for the treatment of PDAC [15]. These studies suggested that PODXL and ITGB1 play important roles in the invasiveness and/or metastasis of PDAC cells, and that increased expression of PODXL and ITGB1 is associated with the poor prognosis of PDAC.

Endoscopic ultrasound-guided fine-needle aspiration biopsy (EUS-FNAB) is widely available and is the gold-standard technique for the preoperative pathological diagnosis of PDAC [16]. If increased expression of PODXL and ITGB1 in PDAC tissues preoperatively obtained by EUS-FNAB is correlated with the postoperative poor prognosis, it may be useful to select which PDAC patients should receive preoperative neoadjuvant therapy to improve their postoperative prognosis. In the present study, we investigated the preoperative use of PODXL and ITGB1 in PDAC tissue samples from EUS-FNAB as useful markers for the prognosis of postoperative PDAC patients in comparison with the TNM staging system. The combination of PODXL with ITGB1 predicted the postoperative outcomes of PDAC patients better than the TNM staging system prior to surgery.

## Results

### PODXL expression in PDAC tissue samples obtained by EUS-FNAB

Immunohistochemical analysis of the expression levels of PODXL and ITGB1 was performed for 24 PDAC tissue samples obtained by EUS-FNAB before resection (Fig 1A), and we used the immunostaining scores of PODXL and ITGB1. The background information of PDAC patients is

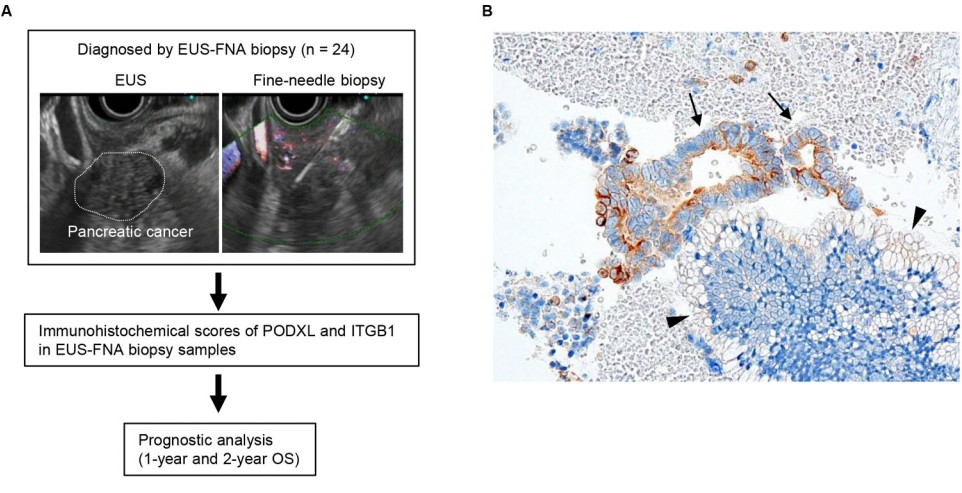

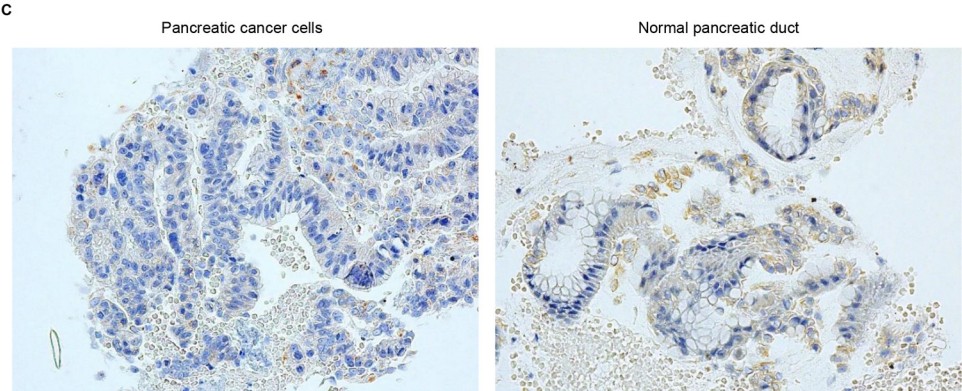

**Fig 1. Expression of PODXL in PDAC tissues obtained by EUS-FNAB.** (A) Study flow chart. (B, C) Representative immunohistochemical staining of PDAC tissue samples using anti-PODXL antibody showing (B) high and (C) low expression of PODXL. Arrows, PDAC cells; arrowheads, normal pancreatic duct epithelium. Magnification: ×400.

shown in Table 1. The tumor diameters were greater than 20 mm for 20/24 PDAC patients, and the numbers of patients with each UICC stage were as follows: IA (n = 5); IB (n = 4); IIA (n = 9); IIB (n = 5); and III (n = 1). Five patients (20.9%) received neoadjuvant therapy consisting of gemcitabine plus oral fluoropyrimidine (S-1) or gemcitabine plus nab-paclitaxel. All PDAC patients were surgically treated, and 18 (75.0%) had R0 and 6 (25.0%) had R1 resections. Seventeen patients (87.5%) received adjuvant chemotherapy consisting of S-1 or gemcitabine.

The immunostaining of PODXL in EUS-FNAB samples is shown in Fig 1B and 1C. PODXL was mainly stained in the cytoplasm and cell membranes of PDAC cells. The scores of immunostaining were classified into a high-expressing PODXL group (score ≥ 4, Fig 1B) and low-expressing PODXL group (≤ 3, Fig 1C), and 13/24 PDAC tissues exhibited strong PODXL staining (Table 1). The intensity of PODXL staining was weak in normal pancreatic ducts of all EUS-FNAB samples (Fig 1B and 1C).

## ITGB1 expression in PDAC tissue samples obtained by EUS-FNAB

ITGB1 was stained in the cytoplasm of PDAC cells in EUS-FNAB samples (Fig 2A and 2B). The scores of immunostaining were classified into a high-expressing ITGB1 group (score ≥ 4,

**Table 1. Summary of characteristics of 24 patients with pancreatic cancer.**

| Characteristics | Percentage (%) | | Charasteristics | Percentage (%) | |
|---|---|---|---|---|---|
| **Age at surgery** | | | **Resection margin status** | | |
| 50–60 | 20.9 | [n = 5] | R0 | 75.0 | [n = 18] |
| 60–70 | 33.3 | [n = 8] | R1 | 25.0 | [n = 6] |
| 70–80 | 33.3 | [n = 8] | R2 | 0 | [n = 0] |
| > 80 | 12.5 | [n = 3] | **CA19-9** | | |
| **Gender** | | | Upregulated | 79.2 | [n = 20] |
| Male | 50.0 | [n = 12] | Normal range | 20.8 | [n = 4] |
| Female | 50.0 | [n = 12] | **Tumor size** | | |
| **Stage*** | | | > = 2.0 cm | 79.2 | [n = 20] |
| IA | 20.8 | [n = 4] | < 2.0cm | 20.8 | [n = 4] |
| IB | 16.7 | [n = 5] | **Neoadjuvant treatment** | | |
| IIA | 37.5 | [n = 9] | Yes | 20.9 | [n = 5] |
| IIB | 20.8 | [n = 5] | No | 79.1 | [n = 19] |
| III | 4.2 | [n = 1] | **Adjuvant treatment** | | |
| **Extent of the tumor*** | | | Yes | 87.5 | [n = 21] |
| T1 | 29.2 | [n = 7] | No | 12.5 | [n = 3] |
| T2 | 33.3 | [n = 8] | **PODXL expression** | | |
| T3 | 37.5 | [n = 9] | Low | 45.8 | [n = 11] |
| T4 | 0 | [n = 0] | High | 54.2 | [n = 13] |
| **Regional lymph nodes*** | | | **ITGB1 expression** | | |
| N0 | 75.0 | [n = 18] | Low | 70.8 | [n = 17] |
| N1 | 25.0 | [n = 6] | High | 29.2 | [n = 7] |
| | | | **PODXL and ITGB1 expression** | | |
| **Distant metastasis*** | | | Others | 79.1 | [n = 19] |
| M0 | 100 | [n = 24] | Both high expression | 20.9 | [n = 5] |
| M1 | 0 | [n = 0] | | | |

*, Classified according to the classification of International Union against Cancer.

Fig 2A) and low-expressing ITGB1 group ($\leq$ 3, Fig 2B), and 7/24 PDAC tissues exhibited strong ITGB1 staining (Table 1). ITGB1 staining was also weak in normal pancreatic ducts of all EUS-FNAB samples (Fig 2A and 2B).

## Association of the overexpression of PODXL and ITGB1 with prognosis

The postoperative overall survival (OS) and relapse-free survival (RFS) rates for PDAC patients using immunostaining scores of PODXL and ITGB1 in EUS-FNAB samples before surgery were calculated by Kaplan-Meier curves. The 1-year and 2-year OS rates of the high-expressing group of PODXL were 0.61 [95% confidence intervals (CI): 0.40–0.94] and 0.46 (95% CI: 0.25–0.83), respectively, and those of the low-expressing group were 1.00 and 0.90 (95% CI: 0.75–1.00) (Fig 3A). The 1-year and 2-year OS rates of the high-expressing group of ITGB1 were 0.42 (95% CI: 0.18–1.00) and 0.28 (95% CI: 0.08–0.92), respectively, and those of the low-expressing group were 0.94 (95% CI: 0.83–1.00) and 0.82 (95% CI: 0.66–1.00) (Fig 3B). The 2-year RFS rate decreased from 0.63 (95% CI: 0.40–0.99) in the low-expressing PODXL group to 0.15 (95% CI: 0.04–0.55) in the high-expressing PODXL group (Fig 4A). The 2-year RFS rate decreased from 0.47 (95% CI: 0.28–0.77) in the low-expressing ITGB1 group to 0.14 (95% CI: 0.02–0.87) in the high-expressing ITGB1 group (Fig 4B). The postoperative OS and RFS

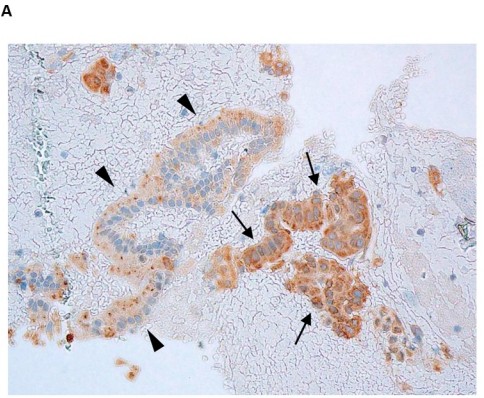

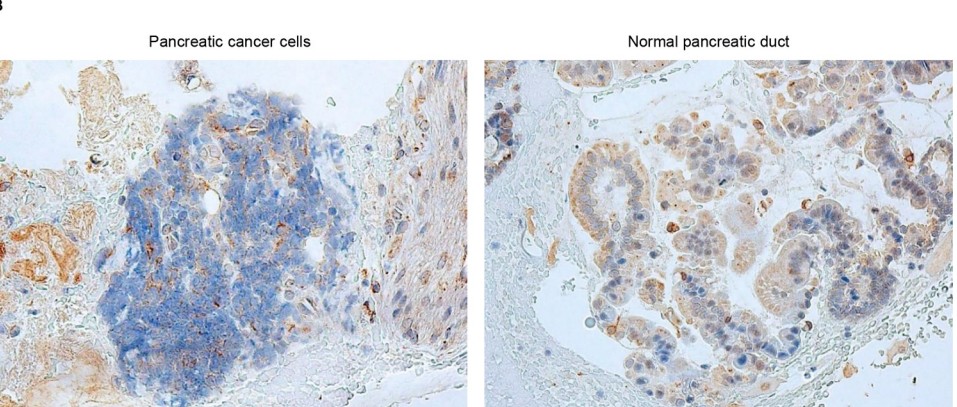

**Fig 2. Expression of ITGB1 in PDAC tissues obtained by EUS-FNAB.** (A, B) Representative immunohistochemical staining of PDAC tissue samples using anti-ITGB1 antibody showing (A) high and (B) low expression of ITGB1. Arrows, PDAC cells; arrowheads, normal pancreatic duct epithelium. Magnification: ×400.

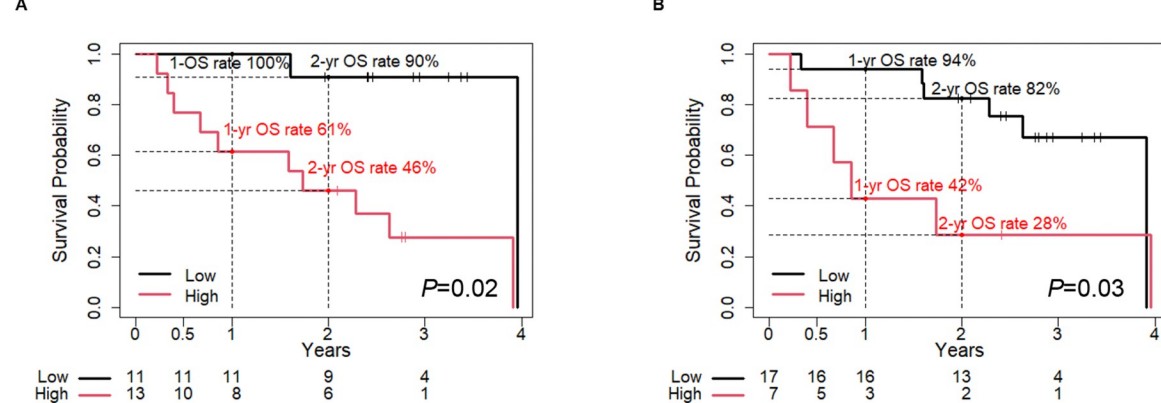

**Fig 3. Correlation between high expression of PODXL and ITGB1 and poor outcomes of PDAC patients.** (A, B) Kaplan-Meier analysis of postoperative OS rates according to (A) PODXL expression and (B) ITGB1 expression in PDAC tissues obtained by EUS-FNAB.

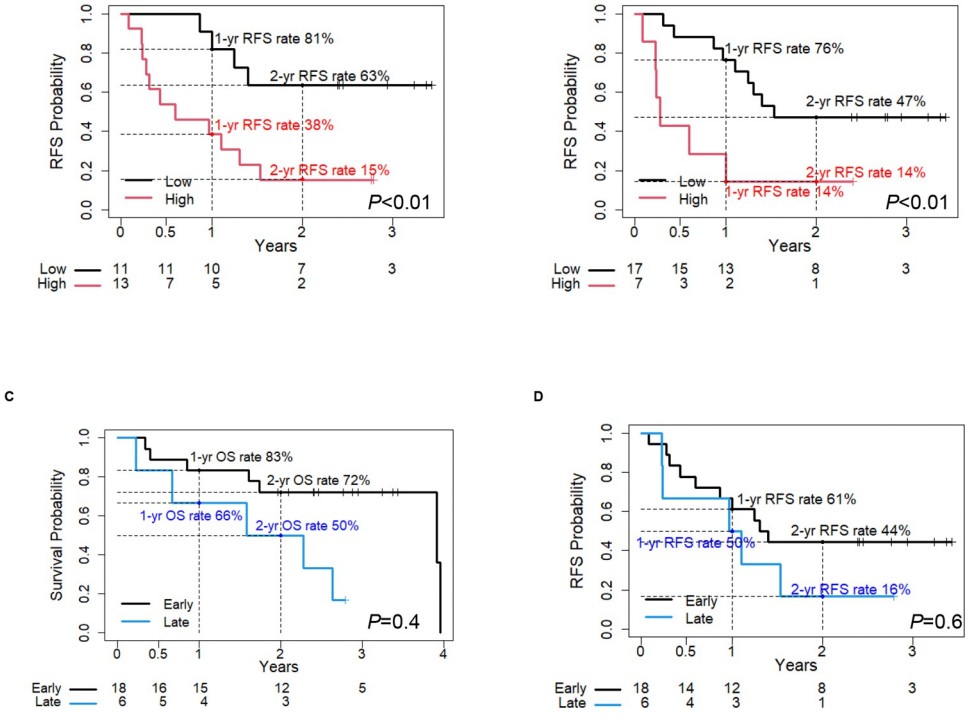

**Fig 4. Relapse-free survival rates of high expression of PODXL and ITGB1, and UICC TNM stage.** (A, B) Kaplan-Meier analysis of postoperative RFS rates according to (A) PODXL expression and (B) ITGB1 expression. (C, D) Kaplan-Meier analysis of postoperative (C) OS rate and (D) RFS rate according to UICC TNM stage (early: UICC stage IA-IIA; late: UICC stage IIB-III).

rates for postoperative PDAC patients according to UICC TNM stage are shown in Fig 4C and 4D. These were consistent with the clinical experience that PDAC patients diagnosed with advanced stage have poorer outcomes than those with an earlier stage.

Univariate Cox regression analysis revealed that three clinicopathological features, including PODXL, UICC TNM stage and lymphatic invasion, are significantly associated with OS ($P<0.05$) (Table 2). Other clinicopathological variables, including resectability, neoadjuvant treatment and adjuvant treatment, were not significant preoperative predictors for prognosis in resected PDAC. Among single clinicopathological variables, the highest hazard ratio (HR) for OS rate was PODXL (12.45, 95% CI: 1.59–97.69), followed by lymphatic invasion (3.78, 95% CI: 1.09–13.23), UICC TNM stage (3.78, 95% CI: 1.09–13.12) and ITGB1 (3.04, 95% CI: 0.89–10.38) (Table 2). In the Japanese guideline of adjuvant chemotherapy for resected PDAC patients, S-1 monotherapy is recommended and gemcitabine monotherapy is used for patients with low tolerability for S-1 [17, 18]. As S-1 or gemcitabine was used for all resected PDAC patients except for those with jaundice or common bile duct stenosis in this study, the HR of adjuvant treatment was unable to be calculated statistically (Table 2). Of note, the HR for the combination of PODXL with ITGB1 was higher (14.37, 95% CI: 3.23–63.99) than that of PODXL and ITGB1. Furthermore, multivariate Cox regression analysis revealed that UICC TNM stage and the combination of PODXL with ITGB1 are significantly associated with OS ($P<0.05$) (Table 2). The combination of PODXL with ITGB1 predicted the prognosis most accurately (HR: 31.16, 95% CI: 4.36–222.45), and HR of UICC TNM stage was 7.36 (95% CI: 1.32–40.93). This suggested that the combination of PODXL with ITGB1 is a useful preoperative predictor of poorer postoperative survival for PDAC. The Akaike information criterion

**Table 2. Univariate and multivariate analysis of prognostic factors for overall survival.**

| | Overall survival | | | |
| --- | --- | --- | --- | --- |
| | Univariate** | | Multivariate*** | |
| | HR (95% CI) | *P* | HR (95% CI) | *P* |
| **Stage*** | | | | |
| IA + IB + IIA | Reference | | Reference | |
| IIB + III | 3.78 (1.09–13.12) | 0.036 | 7.36(1.32–40.93) | 0.023 |
| **Age at surgery** | 1.03 (0.94–1.08) | 0.93 | | |
| **Gender** | | | | |
| Female | Reference | | | |
| Male | 0.58 (0.16–2.05) | 0.39 | | |
| **PODXL expression** | | | | |
| Low | Reference | | | |
| High | 12.45 (1.59–97.69) | 0.017 | | |
| **ITGB1 expression** | | | | |
| Low | Reference | | | |
| High | 3.04 (0.89–10.38) | 0.076 | | |
| **PODXL and ITGB1 expression** | | | | |
| Others | Reference | | Reference | |
| Both high expression | 14.37 (3.23–63.99) | < 0.01 | 31.16(4.36–222.45) | < 0.01 |
| **Extent of the tumor*** | | | | |
| T1 + T2 | Reference | | Reference | |
| T3 + T4 | 3.27 (0.69–15.42) | 0.135 | 2.01(0.87–4.91) | 0.100 |
| **Lymphatic invasion*** | | | | |
| N0 | Reference | | | |
| N1 | 3.78 (1.09–13.23) | 0.036 | | |
| **Resectability** | | | | |
| R0 | Reference | | | |
| R1 | 1.70 (0.49–5.93) | 0.405 | | |
| **Neoadjuvant treatment** | | | | |
| No | Reference | | | |
| Yes | 1.31 (0.34–5.10) | 0.693 | | |
| **Adjuvant treatment** | | | | |
| No | Reference | | | |
| Yes | 0 (0- infinity) | 0.99 | | |
| **CA19-9** | | | | |
| Normal range | Reference | | | |
| Upregulated | 1.49 (0.31–7.07) | 0.617 | | |

*, Classified according to the classification of International Union against Cancer.

**, Univariate analysis was performed for variables including age, sex, UICC TNM stage, PODXL, ITGB1, the combination of PODXL with ITGB1, extent of the tumor, regional lymph nodes, resection margin status, neoadjuvant treatment, adjuvant treatment and CA19-9.

***, Stepwise model selection using the Akaike information criterion (AIC) and multivariate analysis were performed for variables including age, sex, UICC TNM stage, the combination of PODXL with ITGB1, extent of the tumor, regional lymph nodes, resection margin status, neoadjuvant treatment and CA19-9.

(AIC) revealed that other clinicopathological variables, including extent of the tumor, lymphatic invasion, resectability, neoadjuvant treatment and serum cancer antigen 19–9 (CA19-9) concentration, were not able to predict the prognosis of postoperative PDAC patients as accurately as the combination of PODXL with ITGB1. CA19-9 is a tumor marker commonly associated with PDAC.

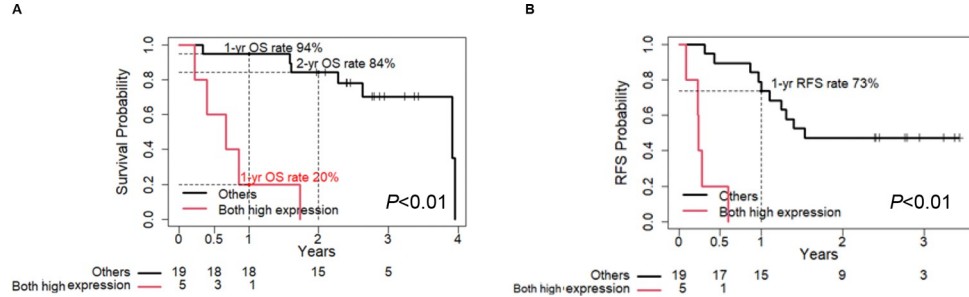

**Fig 5. Correlation between the combination of PODXL with ITGB1 and poor outcomes of PDAC patients.** (A, B) Kaplan-Meier analysis of (A) OS and (B) RFS rates according to the combination of PODXL with ITGB1.

## Ability of the combination of PODXL and ITGB1 to predict the prognosis of PDAC patients

We investigated the potential of using a combination of PODXL and ITGB1 to predict the postoperative prognosis in comparison with PODXL, ITGB1 and UICC TNM stage prior to resection separately. Based on Kaplan-Meier curves, the postoperative OS rate (Fig 5A) and RFS rate (Fig 5B) for PDAC patients with overexpression of both PODXL and ITGB1 (n = 5) in EUS-FNAB samples before surgery were significantly shorter than those for PDAC patients without high expression of both PODXL and ITGB1 (n = 19) (*P*<0.01 for OS rate and RFS rate).

The 1-year and 2-year OS rates of the group highly expressing both PODXL and ITGB1 were 0.20 (95% CI: 0.03–1.00) and 0.00, respectively, and those of the group with low expression of both were 0.94 (95% CI: 0.85–1.00) and 0.84 (95% CI: 0.69–1.00) (Table 3). The OS rates of UICC TNM stage IA-IIA and IIB-III were 0.83 (95% CI: 0.67–1.00) and 0.66 (95% CI: 0.37–1.00) for 1 year, and 0.72 (95% CI: 0.54–0.96) and 0.50 (95% CI: 0.22–1.00) for 2 years, respectively (Table 3). This suggests that the combination of PODXL with ITGB1 is a useful predictor of postoperative outcomes for PDAC patients before resection.

## Ability of the combination of PODXL and ITGB1 to predict the prognosis of PDAC patients with UICC TNM stage IA-IIA

The increasing shift towards neoadjuvant treatments for both resectable and borderline PDAC, and the use of conversion therapy for locally advanced disease suggest the need for biological predictors in addition to the UICC TNM stage [19]. As these predictors are not currently clinically employed, we focused on the ability of the combination of PODXL and ITGB1 to predict the postoperative prognosis of PDAC patients with UICC TNM stage IA-IIA,

**Table 3. Survival rates and median survival times of the combination of PODXL with ITGB1.**

|  | n | Survival rate (95% CI) (%) | | Median survival time (95% CI) (month) |
|---|---|---|---|---|
|  |  | 1-year | 2-year |  |
| **PODXL and ITGB1 expression** |  |  |  |  |
| Others | 19 | 0.94 (0.85–1.00) | 0.84 (0.69–1.00) | 47 (47-NA) |
| Both high expression | 5 | 0.20 (0.03–1.00) | 0.00 (NA-NA) | 8 (5-NA) |
| **UICC TNM stage** |  |  |  |  |
| Stage IA-IIA | 18 | 0.83 (0.67–1.00) | 0.72 (0.54–0.96) | 47 (47-NA) |
| Stage IIB-III | 6 | 0.66 (0.37–1.00) | 0.50 (0.22–1.00) | 23 (8-NA) |

**A**

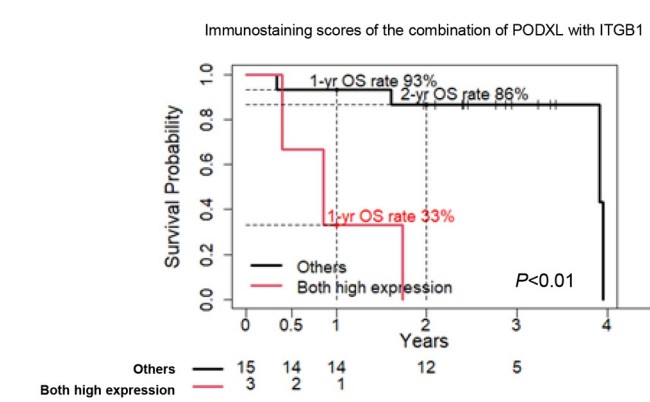

**B**

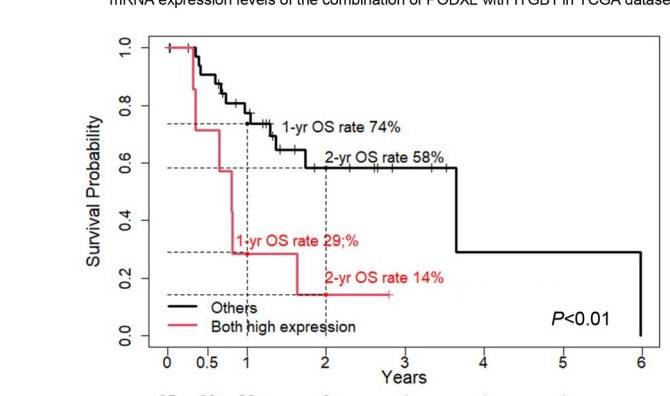

**Fig 6. Correlation between the combination of PODXL with ITGB1 and poor outcomes of PDAC patients at stage IA-IIA.** (A) Kaplan-Meier analysis of OS rate according to the immunohistochemical scores of the combination of PODXL with ITGB1 in PDAC patients at stage IA-IIA. (B) Kaplan-Meier analysis of OS rate according to the mRNA expression levels of the combination of PODXL with ITGB1 in the TCGA set.

including both resectable and borderline PDAC before surgery. Based on Kaplan-Meier curves, the postoperative OS rate for UICC TNM stage IA-IIA PDAC patients with high expression of both PODXL and ITGB1 (n = 3) in EUS-FNAB samples before surgery was significantly shorter than that of UICC TNM stage IA-IIA PDAC patients without high expression of both PODXL and ITGB1 (n = 15) ($P$<0.01) (Fig 6A). The 1-year and 2-year OS rates of UICC TNM stage IA-IIA patients highly expressing both PODXL and ITGB1 compared with other PDAC patients were 0.33 (95% CI: 0.06–1.00) and 0.93 (95% CI: 0.81–1.00), and 0.00 and 0.86 (95% CI: 0.71–1), respectively (Table 4). The prognostic analysis using messenger RNA (mRNA)-seq data of 42 resected UICC TNM stage IA-IIA PDAC samples from The Cancer Genome Atlas (TCGA) database revealed that the postoperative OS rate for PDAC patients with UICC TNM stage IA-IIA with upregulated mRNA levels of both PODXL and ITGB1 (n = 7) was significantly lower than that for PDAC patients without high expression of both PODXL and ITGB1 (n = 35) ($P$ = 0.006) (Fig 6B). This suggests that the combination of

**Table 4. Survival rates and median survival times of the combination of PODXL with ITGB1 at UICC stage IA-IIA.**

| | n | Survival rate (95% CI) (%) | | Median survival time (95% CI) (month) |
|---|---|---|---|---|
| | | **1-year** | **2-year** | |
| **PODXL and ITGB1 expression** | | | | |
| Others | 15 | 0.93 (0.81–1.00) | 0.86 (0.71–1.00) | 47 (47-NA) |
| Both high expression | 3 | 0.33 (0.06–1.00) | 0.00 (NA-NA) | 10 (5-NA) |

PODXL with ITGB1 is useful to predict the postoperative outcomes of PDAC patients with UICC TNM stage IA-IIA prior to surgery.

## Discussion

Preoperative biomarkers were demonstrated to have preliminary value in predicting the prognosis of PDAC patients. Among them, preoperatively high serum levels of both CEA and CA19-9 are associated with a poor postoperative prognosis of resected PDAC patients [20]. Serum KRAS mutations, especially the KRAS G12D mutation, in preoperative cell-free circulating tumor DNA are associated with the poor prognosis of resected PDAC patients [21]. Preoperative serum complement factor B accurately predicts the prognosis of resected PDAC compared with high CEA and CA19-9 [22]. Furthermore, few predictive biomarkers for the prediction of responsiveness to neoadjuvant chemotherapy have been identified in PDAC. A microRNA miRNA-320c, which is associated with the invasiveness of PDAC cells, was reported as a prognostic factor for PDAC to predict the clinical response of gemcitabine [23]. Gemcitabine-based chemotherapy is commonly utilized as a first-line treatment to treat advanced PDAC patients [24]. As radical resection with a negative margin (R0 resection) is the key factor for long-term survival of PDAC [6], potential prognostic factors that are available before resection are necessary for PDAC patients who are commonly considered operable to prepare a neoadjuvant strategy. The present study demonstrated that the high expression of both PODXL and ITGB1 in preoperatively extracted PDAC tissues obtained by EUS-FNAB is closely associated with the poor prognosis of PDAC patients after resection ($P<0.01$), even though there was no notable difference in prognosis between R0 resection and R1 resection ($P = 0.405$).

PODXL, which binds to the cytoskeletal protein gelsolin, promotes PDAC cell invasion by increasing membrane protrusions with abundant peripheral actin structures [11]. A retrospective clinical study revealed that serum PODXL can significantly distinguish PDAC patients with UICC stage 0/I/II from control individuals compared with serum CA19-9 [25]. The difficulty of detecting the presence of localized retroperitoneal invasion and micro-metastasis in PDAC patients by diagnostic imaging is the main reason for the high rate of local recurrence and/or distant metastasis after resection [6, 26]. The recurrence rates after curative surgery for PDAC are 56.7, 76.6 and 84.1% at 1, 2 and 5 years, respectively [27]. Over 90% of resected PDAC patients develop recurrence in the abdominal cavity [28]. The immunohistochemically high expression of both PODXL and ITGB1 in resected early-stage (UICC TNM stage 0-IIA) PDAC tissues is associated with a poorer prognosis [12], suggesting that PODXL and ITGB1 play important roles in the invasiveness and early recurrence of PDAC, and this correlates with the function of PODXL in promoting the invasiveness of PDAC cells in *in vitro* experiments [11]. In this study, the combination of PODXL with ITGB1 most accurately predicted the postoperative outcomes of PDAC patients before resection according to univariate and multivariate analyses. Importantly, it is possible that the combination of PODXL with ITGB1 is a useful predictor of postoperative outcomes for PDAC patients at an earlier stage before resection.

Evidence of improved OS with neoadjuvant therapy for borderline resectable or locally advanced PDAC is supported by results from large cancer databases and meta-analyses of non-randomized trials, and a meta-analysis of non-randomized cohorts suggested the utility of FOLFIRINOX (5-fluorouracil, leucovorin, irinotecan and oxaliplatin), which is currently used many institutions [29]. However, as clinical evidence from randomized phase III trials using neoadjuvant therapy for borderline resectable or locally advanced PDAC is limited [8, 27], guidelines for its use are not well defined. The most effective regimens, FOLFIRINOX or gemcitabine plus nab-paclitaxel, have been expanded into neoadjuvant treatments for resectable PDAC because of their potential benefits, including for the early treatment of occult micro-metastasis [30, 31]. A randomized phase II/III trial (Prep-02/JSAP05) performed in Japan demonstrated that gemcitabine plus S-1 with upfront surgery improves the postoperative overall survival of resectable PDAC patients [32]. In the present study, 5 patients (20.9%) received neoadjuvant chemotherapy before surgery, and there was no notable difference in prognosis between resected PDAC patients with neoadjuvant chemotherapy and resected PDAC patients without neoadjuvant chemotherapy ($P$ = 0.693). The development of neoadjuvant therapeutic approaches that are more beneficial than FOLFIRINOX, gemcitabine plus nab-paclitaxel or gemcitabine plus S-1 is important to increase survival compared with upfront surgery.

As no reliable biomarkers can gauge the response to neoadjuvant therapy prior to the initiation of treatment [33], it is difficult to discriminate patients with operative PDAC in whom neoadjuvant therapy may be effective and suitable. Most patients received upfront surgery without neoadjuvant therapy in this study. Therefore, PDAC patients with the overexpression of PODXL and ITGB1 need to be scrutinized closely for incomplete resectability or operability if patients with early-stage PDAC do not receive neoadjuvant chemotherapy. Further studies are needed to determine whether prognostic predictors, including PODXL and ITGB1, can be used to optimize the selection of PDAC patients who will benefit from neoadjuvant treatment and eventually improve outcomes. Toward this end, we started a prospective clinical study (UMIN000034022) to clarify the association of the overexpression of PODXL and ITGB1 with the benefit of neoadjuvant therapies in resectable and borderline resectable PDAC patients.

In conclusion, immunohistochemical scores of PODXL and ITGB1 in preoperative EUS-FNAB samples accurately predicted the postoperative prognosis of PDAC patients better than the UICC TNM stage. The combination of PODXL with ITGB1 can discriminate PDAC patients with a poorer prognosis who have early-stage PDAC prior to surgery. Patients with PDAC that overexpresses both PODXL and ITGB1 should be considered for neoadjuvant therapy first instead of undergoing upfront surgery. However, the use of PODXL and ITGB1 overexpression to optimize neoadjuvant therapy for resectable PDAC patients requires further testing in prospective studies.

## Materials and methods

### Primary human PDAC samples

We retrospectively enrolled 24 PDAC patients who were histologically diagnosed with PDAC from tissue samples obtained by EUS-FNAB before resection, and having UICC TNM stage 0-III tumors without distant metastasis between March 2015 and October 2017 at Kochi Medical School Hospital and Kanagawa Cancer Center. These patients were classified into three resectability groups: resectable, borderline resectable or locally unresectable, according to the 7th edition of the General Rules for the Study of Pancreatic Cancer by the Japan Pancreas Society [34]. Five of 24 patients received neoadjuvant chemotherapy with gemcitabine plus S-1 or gemcitabine plus nab-paclitaxel before resection of PDAC. All patients underwent resection,

and 17 of 24 patients received adjuvant chemotherapy with S-1 or gemcitabine after the resection of PDAC. Pre-treatment serum CA19-9 was measured at certified laboratories associated with the hospital where the patients were treated. Postoperative follow-up methods and acquisition of the medical records were described previously [12]. Observation was censored at PDAC-related death or at the end of observation at least 2 years after resection.

This study was registered in UMIN-CTR (UMIN000032835) and approved by the ethics review boards of Kochi Medical School (approval number: ERB-104012) and Kanagawa Cancer Center (approval number: 2018–131). Obtaining written informed consent from PDAC patients was waived because of the retrospective and observational nature of the analyses, and the opt-out method was approved by the ethics review boards of Kochi Medical School and Kanagawa Cancer Center. The information regarding this study was provided to patients through the institutional website of Kochi Medical School and Kanagawa Cancer Center for obtaining consent. PDAC patients who did not want to participate in this study were able to request to opt-out to prevent trial enrollment.

## Immunohistochemical staining

Immunohistochemistry was performed as described previously [11, 35]. The staining intensity of PODXL and ITGB1 in PDAC cells was scored and compared with the normal pancreatic duct epithelium, as described previously [12]. The expression levels were classified into low and high groups based on the score with reference to previous reports [12, 36].

## Statistical analysis

All statistical analyses were performed using R (version 3.3.3; The R Foundation, Wien, Austria) with the packages "KMsurv", "rms" and "survival" as described previously [37]. The survival time was from the date of EUS-FNAB and the analysis was timed to PDAC-related death. We used arbitrary UICC stage categories (IA, IB, IIA vs. IIB and III) and extent of the tumor (T1 and T2 vs. T3 and T4). Estimates of survival probabilities were performed by the Kaplan-Meier method and evaluated by the Gehan-Wilcoxon test. Univariate and multivariate analyses for the chosen explanatory variables were performed by the Cox proportional hazards model to estimate the HR with 95% CI, and $P$-values were calculated by the z test. The Efron parameter approach method was used in the Cox proportional hazards model. Univariate analysis was performed for variables including age, sex, UICC TNM stage, PODXL, ITGB1, the combination of PODXL with ITGB1, extent of the tumor, regional lymph nodes, resection margin status, neoadjuvant treatment, adjuvant treatment and CA19-9. We were unable to build a multi-variate regression model including all the variables due to the risk of overfitting. Instead, forward-backward stepwise model selection using the AIC and multivariate analysis was performed for variables including age, sex, UICC TNM stage, the combination of PODXL with ITGB1, extent of the tumor, regional lymph nodes, resection margin status, neoadjuvant treatment and CA19-9. All statistical tests were two-tailed and $p<0.05$ was considered significant.

## Prognostic analysis using the public PDAC gene expression profile data

TCGA data were retrieved from UCSC Xena [38]. The clinical data obtained from 196 resected PDAC patients and RNA-seq expression data obtained from 183 resected PDAC patients were analyzed. Expression data included the gene-level transcription estimates, as in $\log_2(x+1)$ transformed RNA-Seq by Expectation-Maximization (RSEM) normalized count (cBioinformatics Co., LTD., Tokyo, Japan). We obtained expression data from 42 resected PDAC patients with UICC TNM stage IA, IB and IIA. These 42 patients were divided into two groups

by PODXL and ITGB1 expression in the PDAC biopsy samples obtained prior to surgery. The cut-off value was exploratorily identified using percentile values referring to the ratio of UICC TNM stage IA-IIA PDAC patients without high expression of PODXL or ITGB1 in the immunohistochemical analysis of this study (11/18 for PODXL and 13/18 for ITGB1). The 61st percentile for PODXL and 72nd percentile for ITGB1 were used to analyze the TCGA dataset. We defined patients with high expression of both genes as the "Both high expression group". Other patients were defined as the "Other group". OS rates of "Both high expression group" and "Other group" were calculated according to the Kaplan-Meier method and evaluated by the Gehan-Wilcoxon test. Regardless of the statistical test performed, differences with $p<0.05$ were considered significant.

## Acknowledgments

We thank Yuri Jobu, Chiharu Tanaka, Miki Nishigawa, Hitomi Seki and Shunichi Manabe for their excellent technical assistance.

## Author Contributions

**Conceptualization:** Keisuke Taniuchi.

**Data curation:** Keisuke Taniuchi, Makoto Ueno, Tomoyuki Yokose, Masahiko Sakaguchi.

**Formal analysis:** Masahiko Sakaguchi, Mitsunari Ogasawara, Seiji Naganuma.

**Funding acquisition:** Keisuke Taniuchi.

**Investigation:** Makoto Ueno.

**Methodology:** Keisuke Taniuchi, Masahiko Sakaguchi, Seiji Naganuma.

**Project administration:** Keisuke Taniuchi.

**Resources:** Keisuke Taniuchi, Tomoyuki Yokose.

**Software:** Masahiko Sakaguchi.

**Supervision:** Takuhiro Kosaki, Mutsuo Furihata.

**Validation:** Reiko Yoshioka, Mitsunari Ogasawara, Seiji Naganuma, Mutsuo Furihata.

**Writing – original draft:** Keisuke Taniuchi.

**Writing – review & editing:** Mutsuo Furihata.

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
