## [Decision Letter · Decision Letter 0]

8 Jul 2021

PONE-D-21-14241

Upregulation of PODXL and ITGB1 in pancreatic cancer tissues preoperatively obtained by EUS-FNAB correlates with unfavorable prognosis of postoperative pancreatic cancer patients

PLOS ONE

Dear Dr. Taniuchi,

Thank you for submitting your manuscript to PLOS ONE. After careful consideration, we feel that it has merit but does not fully meet PLOS ONE’s publication criteria as it currently stands. Therefore, we invite you to submit a revised version of the manuscript that addresses the points raised below during the review process by the reviewers.

We look forward to receiving your revised manuscript.

Kind regards,

Surinder K. Batra

Academic Editor

PLOS ONE

Journal Requirements:

2. Please provide additional information about the opt out mechanism employed in your study. In your ethics statement in the manuscript and in the online submission form, please ensure that you have discussed whether all data/samples were fully anonymized before you accessed them and/or whether the IRB or ethics committee waived the requirement for informed consent. If patients provided informed written consent to have data/samples from their medical records used in research, please include this information.

[We thank Yuri Jobu, Chiharu Tanaka, Miki Nishigawa, Hitomi Seki,and Shunichi Manabe fortheir excellent technical assistance.This study was supported by Grants-in-Aid for Scientific Research (KAKENHI: 17K09463, 19K08446, and 20K07699), and AMED under Grant Number JP19lm0203007.]

 [Grants-in-Aid for Scientific Research (KAKENHI): 17K09463 (kt), 19K08446 (kt), and 20K07699 (mo)]

4. We noticed you have some minor occurrence of overlapping text with the following previous publication(s), which needs to be addressed:

: Taniuchi K, Furihata M, Naganuma S, Sakaguchi M, Saibara T (2019) Overexpression of PODXL/ITGB1 and BCL7B/ITGB1 accurately predicts unfavorable prognosis compared to the TNM staging system in postoperative pancreatic cancer patients. PLoS ONE 14(6): e0217920. https://doi.org/10.1371/journal.pone.0217920

In your revision ensure you cite all your sources (including your own works), and quote or rephrase any duplicated text outside the methods section. Further consideration is dependent on these concerns being addressed.

Reviewers' comments:

Reviewer's Responses to Questions

**Comments to the Author**

1. Is the manuscript technically sound, and do the data support the conclusions?

Reviewer #1: Partly

Reviewer #2: Partly

2. Has the statistical analysis been performed appropriately and rigorously? 

Reviewer #1: Yes

Reviewer #2: No

3. Have the authors made all data underlying the findings in their manuscript fully available?

Reviewer #1: Yes

Reviewer #2: Yes

4. Is the manuscript presented in an intelligible fashion and written in standard English?

Reviewer #1: No

Reviewer #2: Yes

5. Review Comments to the Author

Reviewer #1: The authors in the manuscript entitled ‘Upregulation of PODXL and ITGB1 in pancreatic cancer tissues preoperatively obtained by EUS-FNAB correlates with unfavorable prognosis of postoperative pancreatic cancer patients’ investigated the expressions of PODXL and ITGB1 as preoperative markers for extrapolating the postoperative prognosis of pancreatic cancer patients. The authors claim that prior to surgery, a higher expression of PODXL and ITGB1 significantly predicted the postoperative outcomes of pancreatic cancer patients as compared to the TNM staging system. The investigators also suggest that the expression status of PODXL and ITGB1 will direct the patient stratification to receive preoperative neoadjuvant therapy to improve their postoperative prognosis. The overall idea of the work is interesting and seems promising to benefit the pancreatic cancer biomarker field of research, and further improve the patient prognostication. However, there are some major concerns that need to be addressed before considering this manuscript for publication:

1. The rationale of selecting these two markers for a “directed approach” of biomarker selection wasn’t very clear from the introduction and discussion parts. With the accumulating evidence in biomarker research over the past few years, several markers have been studied of which many showed promise in pilot studies but eventually failed to recapitulate to the clinics. Hence, it is essential to have a strong rationale in terms of functional requirement of the protein (and not merely the expression) during the particular cancer stage under investigation.

2. In Table 4, the authors have compared the survival kinetics of TNM stage IA-IIA patients with high expression of PODXL and ITGB1 and those with low expression of both these markers. The number of patients in the high expressor category is too less (n=3), while those on the other arm is n=15. It is understandable that it is a longitudinal study, hence patient recruitment might be difficult. The authors should try to increase the patient number in the first category, or at least try to derive transcriptomic analysis from the patient information available in TCGA or other bioinformatics platforms. This will support their claims.

3. The authors at multiple places claimed that the PODXL and ITGB1 can independently perform as pre-operative predictors, better than TNM staging. However, I don’t see any analysis where the authors do a blinded study on the survival statistics of PDAC patients based on the expression of PODXL and ITGB1, and then look for their TNM stage. The authors segregated patients in stage IA-IIA, and then correlated their survival with expression of these markers. A blinded study on a validation cohort will be helpful, even if it is on a small group of patients.

4. There are some grammar and typographical mistakes throughout the text. Authors are encouraged to correct them in the revised manuscript after a professional English editing.

5. The manuscript has some tables with elaborate information and logistics. The authors are encouraged to incorporate table legends or footnotes, explaining the critical terminologies and providing a statement on the overall observation from the analysis.

Reviewer #2: In their article “Upregulation of PODXL and ITGB1 in pancreatic cancer tissues preoperatively

obtained by EUS-FNAB correlates with unfavorable prognosis of postoperative

pancreatic cancer patients”. Taniuchi and colleagues present there working regarding the ability of PODXL and ITGB1 immunostaining in EUS biopsies to predict post-operative prognosis. Overall, the work is interesting and well written. The samples included, while small in number, are appropriate for this investigation. Moreover, the scientific framing of the article, in the intro and discussion sections, makes the potential clinical relevance of this study clear. Finally, the author’s assertions regarding the utility of these findings are, in this reviewer’s opinion, well-grounded and not overstated. There are, however, several points for further consideration that may further enhance the readability and strength the author’s claims.

Minor Comments

1. Image quality for figures 1 and 2 is limited and makes assessment of the IHC quality and histology more difficult

2. Font in Figures 3-6 are difficult to read. Clarity would be enhanced by improving the legibility in these figures.

3. Within the statistical analysis section of the methods and materials, it is important to state which factors were used in the generation of the cox proportional hazard models. The inclusion/exclusion of various patient and disease factors can have marked impacts on the assessment of single factor as being a

4. The lack of p-values present in the figures as well as the text manuscript and figure legends make it slightly unwieldy for the reader to easily assess the statistical significance of the various comparisons present in this study

5. Please indicate the statistic used for calculating univariate survival. There are numerous different statistics that can be used within the overall framework of Kaplan-Meier analysis. These varying statistics weigh early and late event differently and can have notable effects on the outcomes of such analyses.

6. In table 2 under the univariate column, the HR for stage is listed as 3.78 with a 95% confidence interval that does not include 1, yet the p-values is not significant. By definition, this is significant. Please explain.

Major Comments

1. The inclusion of only PODXL and ITGB1 staining profiles in multivariate models while excluding/ not reporting the results of other significant univariate factors (such as Lymphatic invasion or the very closely numerically related factor of stage) seems to be an inappropriate use of cox proportional hazards. These other factors must be included in the model and the results reported for such an analysis; without this the findings are difficult to interpret and may be misleading.

2. Given that this study is conducted in a subset of patients who have undergone surgery, this reviewer feels that it is imperative to report the R-status (R0, R1, R2) of the resection. Moreover, this reviewer feels that this will be an important factor to include in the univariate survival analysis and the multivariate analysis if significant due to the fact that differences in the R value can have a major impact on survival independent of the staging. Previous literature supports this notion indicating that patients who undergo an R2 resection do not have improved survival over patients with unresectable disease.

3. Similarly, it is odd that the adjuvant and neoadjuvant chemotherapy were not analyzed in univariate survival analyses. This reviewer understands that in the previous paper these authors did not find that adjuvant therapy and a significant effect on survival. Nonetheless, the authors mention that well controlled trials have demonstrated benefits to adjuvant treatments. Consequently, this reviewer feels that it is important to address the adjuvant and neo adjuvant therapy in the univariate survival analysis and multivariate analyses should it prove to be a significant factor.

6. PLOS authors have the option to publish the peer review history of their article (what does this mean?). If published, this will include your full peer review and any attached files.

Reviewer #1: No

Reviewer #2: No

---

## [Author Response · Author response to Decision Letter 0]

21 Aug 2021

Dr. Surinder Batra,

August 19, 2021

Submission ID: PONE-D-21-14241

Upregulation of PODXL and ITGB1 in pancreatic cancer tissues preoperatively obtained by EUS-FNAB correlates with unfavorable prognosis of postoperative pancreatic cancer patients

Dear Dr. Surinder Batra,

I am returning the above manuscript that was revised according to your letter dated the 8th of July. Our responses to the reviewers’ comments are provided below. 

Response : The manuscript was revised according to PLOS ONE's style requirements, including those for file naming.

2. Please provide additional information about the opt out mechanism employed in your study. In your ethics statement in the manuscript and in the online submission form, please ensure that you have discussed whether all data/samples were fully anonymized before you accessed them and/or whether the IRB or ethics committee waived the requirement for informed consent. If patients provided informed written consent to have data/samples from their medical records used in research, please include this information.

Response : Additional information about the opt-out system was added to the section of “Primary human PDAC samples” of the Materials and Methods section in the revised manuscript.

[We thank Yuri Jobu, Chiharu Tanaka, Miki Nishigawa, Hitomi Seki,and Shunichi Manabe fortheir excellent technical assistance. This study was supported by Grants-in-Aid for Scientific Research (KAKENHI: 17K09463, 19K08446, and 20K07699), and AMED under Grant Number JP19lm0203007.]

 [Grants-in-Aid for Scientific Research (KAKENHI): 17K09463 (kt), 19K08446 (kt), and 20K07699 (mo)]

Response: The funding-related text was removed from the revised manuscript. The Funding Statement is as follows: Grants-in-Aid for Scientific Research (KAKENHI): 17K09463 (kt), 19K08446 (kt), 19K07461 (mf), 20K07699 (mo), 20K07806 (ry) and 20K08359 (kt), and Japan Agency for Medical Research and Development (AMED) under Grant Number JP19lm0203007 (kt).

4. We noticed you have some minor occurrence of overlapping text with the following previous publication(s), which needs to be addressed: Taniuchi K, Furihata M, Naganuma S, Sakaguchi M, Saibara T (2019) Overexpression of PODXL/ITGB1 and BCL7B/ITGB1 accurately predicts unfavorable prognosis compared to the TNM staging system in postoperative pancreatic cancer patients. PLoS ONE 14(6): e0217920. https://doi.org/10.1371/journal.pone.0217920.

In your revision ensure you cite all your sources (including your own works), and quote or rephrase any duplicated text outside the methods section. Further consideration is dependent on these concerns being addressed.

Response: The overlapping text with the previous publication (PLoS ONE 14(6): e0217920) was addressed in the revised manuscript.

Reviewer #1:

We appreciate the Reviewer’s useful comments.

Comments:

1. The rationale of selecting these two markers for a “directed approach” of biomarker selection wasn’t very clear from the introduction and discussion parts. With the accumulating evidence in biomarker research over the past few years, several markers have been studied of which many showed promise in pilot studies but eventually failed to recapitulate to the clinics. Hence, it is essential to have a strong rationale in terms of functional requirement of the protein (and not merely the expression) during the particular cancer stage under investigation.

Response: In response to the Reviewer’s comments, the Abstract section, 2nd paragraph of the Introduction section, and the 1st and 2nd paragraphs of the Discussion section were revised. 

2. In Table 4, the authors have compared the survival kinetics of TNM stage IA-IIA patients with high expression of PODXL and ITGB1 and those with low expression of both these markers. The number of patients in the high expressor category is too less (n=3), while those on the other arm is n=15. It is understandable that it is a longitudinal study, hence patient recruitment might be difficult. The authors should try to increase the patient number in the first category, or at least try to derive transcriptomic analysis from the patient information available in TCGA or other bioinformatics platforms. This will support their claims.

Response: In response to the Reviewer’s comments, the prognostic analysis of TNM stage IA-IIA patients with high expression of PODXL and ITGB1 using the TCGA bioinformatics platform was added in Fig. 6B.

3. The authors at multiple places claimed that the PODXL and ITGB1 can independently perform as pre-operative predictors, better than TNM staging. However, I don’t see any analysis where the authors do a blinded study on the survival statistics of PDAC patients based on the expression of PODXL and ITGB1, and then look for their TNM stage. The authors segregated patients in stage IA-IIA, and then correlated their survival with expression of these markers. A blinded study on a validation cohort will be helpful, even if it is on a small group of patients.

Response: As the Reviewer mentioned, a blinded study on the survival statistics of pancreatic cancer patients based on the expression of PODXL and ITGB1 was not performed. To better clarify this, the sentences about the results of multivariate Cox regression analysis in “Associations of overexpression of PODXL and ITGB1 with prognosis” in the Results were revised in the revised manuscript. Concerning the prognostic analysis of TNM stage IA-IIA patients with high expression of PODXL and ITGB1, the results of the TCGA bioinformatics platform were added to Fig. 6B.

4. There are some grammar and typographical mistakes throughout the text. Authors are encouraged to correct them in the revised manuscript after a professional English editing.

Response: Grammar mistakes were corrected in the revised manuscript.

5. The manuscript has some tables with elaborate information and logistics. The authors are encouraged to incorporate table legends or footnotes, explaining the critical terminologies and providing a statement on the overall observation from the analysis.

Response: In response to the Reviewer’s comments, table footnotes were incorporated into Table 2.

Reviewer #2:

We appreciate the Reviewer’s useful comments.

1. Image quality for Figures 1 and 2 is limited and makes assessment of the IHC quality and histology more difficult.

Response: The images of Figures 1 and 2 were exchanged, which improved the quality of the revised manuscript.

2. Font in Figures 3-6 are difficult to read. Clarity would be enhanced by improving the legibility in these figures.

Response: Font size in Figures 3-6 was enlarged in the revised manuscript.

3. Within the statistical analysis section of the methods and materials, it is important to state which factors were used in the generation of the cox proportional hazard models. The inclusion/exclusion of various patient and disease factors can have marked impacts on the assessment of single factor as being a

Response: In response to the Reviewer’s comments, information about variables for the univariate and multivariate analysis were added to the statistical analysis section of the Methods and Materials and footnote in Table 2.

4. The lack of p-values present in the figures as well as the text manuscript and figure legends make it slightly unwieldy for the reader to easily assess the statistical significance of the various comparisons present in this study.

Response: In response to the Reviewer’s comments, the p-values for the Kaplan-Meier survival analysis were added to Figures 3-6, and p-values for the univariate and multivariate analysis were added in the text and Table 2. Stepwise model selection from the variables including age, sex, UICC TNM stage, extent of the tumor, regional lymph nodes, resection margin status, neoadjuvant treatment, CA19-9, and the combination of PODXL with ITGB1 using the Akaike information criterion (AIC) was performed in multivariate analysis; therefore, the hazard ratio and p-values of significant variables were added to Table 2.

5. Please indicate the statistic used for calculating univariate survival. There are numerous different statistics that can be used within the overall framework of Kaplan-Meier analysis. These varying statistics weigh early and late event differently and can have notable effects on the outcomes of such analyses.

Response: In response to the Reviewer’s comments, information about the statistical methods used in the univariate analysis and Kaplan-Meier survival analysis was added to “Statistical analysis” in the Materials and Methods section.

6. In table 2 under the univariate column, the HR for stage is listed as 3.78 with a 95% confidence interval that does not include 1, yet the p-values is not significant. By definition, this is significant. Please explain.

Response: The erroneous p-value for TNM stage was corrected in Table 2 of the revised manuscript.

Major Comments

1. The inclusion of only PODXL and ITGB1 staining profiles in multivariate models while excluding/ not reporting the results of other significant univariate factors (such as Lymphatic invasion or the very closely numerically related factor of stage) seems to be an inappropriate use of cox proportional hazards. These other factors must be included in the model and the results reported for such an analysis; without this the findings are difficult to interpret and may be misleading.

Response: Information about variables for the multivariate analysis was added to the statistical analysis section of the Methods and Materials and footnote in Table 2. In the multivariate analysis, variables included age, sex, UICC TNM stage, the combination of PODXL with ITGB1, extent of the tumor, regional lymph nodes, resection margin status, neoadjuvant treatment and CA19-9, and stepwise model selection using the Akaike information criterion and multivariate analysis were performed.

2. Given that this study is conducted in a subset of patients who have undergone surgery, this reviewer feels that it is imperative to report the R-status (R0, R1, R2) of the resection. Moreover, this reviewer feels that this will be an important factor to include in the univariate survival analysis and the multivariate analysis if significant due to the fact that differences in the R value can have a major impact on survival independent of the staging. Previous literature supports this notion indicating that patients who undergo an R2 resection do not have improved survival over patients with unresectable disease.

Response: Information about the resection margin status (R-status) was added to Table 1-2, and the R-status was included as one of the variables in the univariate and multivariate analyses. R-status failed to predict the prognosis of postoperative pancreatic cancer patients accurately compared with the combination of PODXL with ITGB1. This result is described in “Associations of overexpression of PODXL and ITGB1 with prognosis” in the Results section of the revised manuscript.

3. Similarly, it is odd that the adjuvant and neoadjuvant chemotherapy were not analyzed in univariate survival analyses. This reviewer understands that in the previous paper these authors did not find that adjuvant therapy and a significant effect on survival. Nonetheless, the authors mention that well controlled trials have demonstrated benefits to adjuvant treatments. Consequently, this reviewer feels that it is important to address the adjuvant and neo adjuvant therapy in the univariate survival analysis and multivariate analyses should it prove to be a significant factor.

Response: The adjuvant and neoadjuvant treatments were included as variables for the univariate analysis. As S-1 or gemcitabine was used for all resected PDAC patients except for those with jaundice or common bile duct stenosis in this study, the hazard ratio of adjuvant treatment was unable to be calculated statistically, as described in Table 2. The adjuvant and neoadjuvant treatments were not associated with the prognosis of postoperative pancreatic cancer patients compared with the combination of PODXL with ITGB1. This suggests that the development of neoadjuvant therapeutic approaches that are more beneficial than currently available chemotherapy regimens, such as FOLFIRINOX, gemcitabine plus nab-paclitaxel or gemcitabine plus S-1, is important to improve survival.

I hope that the revised manuscript is now acceptable for publication.

Yours sincerely,

Keisuke Taniuchi, MD, PhD.

Department of Gastroenterology and Hepatology

Kochi Medical School, Kochi University

Kohasu, Oko-cho, Nankoku, Kochi 783-8505, Japan 

Phone: +81-88-880- 2338; Fax: +81-88-880- 2338

Email: ktaniuchi@kochi-u.ac.jp

---

## [Decision Letter · Decision Letter 1]

9 Nov 2021

PONE-D-21-14241R1Upregulation of PODXL and ITGB1 in pancreatic cancer tissues preoperatively obtained by EUS-FNAB correlates with unfavorable prognosis of postoperative pancreatic cancer patientsPLOS ONE

Dear Dr. Taniuchi,

Thank you for submitting your manuscript to PLOS ONE. After careful consideration, we feel that it has merit but does not fully meet PLOS ONE’s publication criteria as it currently stands. Therefore, we invite you to submit a revised version of the manuscript that addresses the points raised below during the review process by one of the reviewers. Specially, re-analysis of the TCGA data based on controls, PDAC and neuroendocine tumors.

We look forward to receiving your revised manuscript.

Kind regards,

Surinder K. Batra

Academic Editor

PLOS ONE

Journal Requirements:

Reviewers' comments:

Reviewer's Responses to Questions

**Comments to the Author**

1. If the authors have adequately addressed your comments raised in a previous round of review and you feel that this manuscript is now acceptable for publication, you may indicate that here to bypass the “Comments to the Author” section, enter your conflict of interest statement in the “Confidential to Editor” section, and submit your "Accept" recommendation.

Reviewer #1: All comments have been addressed

Reviewer #2: (No Response)

2. Is the manuscript technically sound, and do the data support the conclusions?

Reviewer #1: Yes

Reviewer #2: Yes

3. Has the statistical analysis been performed appropriately and rigorously? 

Reviewer #1: Yes

Reviewer #2: Yes

4. Have the authors made all data underlying the findings in their manuscript fully available?

Reviewer #1: Yes

Reviewer #2: Yes

5. Is the manuscript presented in an intelligible fashion and written in standard English?

Reviewer #1: Yes

Reviewer #2: Yes

6. Review Comments to the Author

Reviewer #1: (No Response)

Reviewer #2: This reviewer appreciates the the changes that the author's made to the previous version of the manuscript. These changes have improved the quality and significance of the paper.

However, the use of TCGA data, while a meaningful addition, is not executed appropriately in this manuscript. In the TCGA PAAD dataset, there a number of benign and malignant entities which comprise the the 183 samples in the full data set. To indicate that these are all PDAC samples is misleading and incorrect. Moreover the inclusion of pancreatic neuroendocrine tumors (included in the 183 samples) in the analysis has the potential to bias or confound the meaning of outcomes studies. Please repeat this analysis using only the samples in the TCGA PAAD dataset that are confirmed to be PDAC.

7. PLOS authors have the option to publish the peer review history of their article (what does this mean?). If published, this will include your full peer review and any attached files.

Reviewer #1: No

Reviewer #2: No

---

## [Author Response · Author response to Decision Letter 1]

7 Jan 2022

Comments:

1. However, the use of TCGA data, while a meaningful addition, is not executed appropriately in this manuscript. In the TCGA PAAD dataset, there a number of benign and malignant entities which comprise the the 183 samples in the full data set. To indicate that these are all PDAC samples is misleading and incorrect. Moreover, the inclusion of pancreatic neuroendocrine tumors (included in the 183 samples) in the analysis has the potential to bias or confound the meaning of outcomes studies. Please repeat this analysis using only the samples in the TCGA PAAD dataset that are confirmed to be PDAC.

Response:

In response to the Reviewer’s comments, we excluded neuroendocrine tumors and repeated the prognostic analysis of 42 PDACs that were confirmed to be TNM stage IA-IIA using the TCGA bioinformatics platform. Based on Kaplan-Meier curves, the postoperative overall survival rate for PDAC patients with UICC TNM stage IA-IIA with upregulated mRNA levels of both PODXL and ITGB1 (n = 7) was significantly lower than that for PDAC patients without high expression of both PODXL and ITGB1 (n = 35) (P=0.006), as shown in Fig. 6B.

---

## [Decision Letter · Decision Letter 2]

28 Feb 2022

Upregulation of PODXL and ITGB1 in pancreatic cancer tissues preoperatively obtained by EUS-FNAB correlates with unfavorable prognosis of postoperative pancreatic cancer patients

PONE-D-21-14241R2

Dear Dr. Taniuchi,

We’re pleased to inform you that your manuscript has been judged scientifically suitable for publication and will be formally accepted for publication once it meets all outstanding technical requirements.

Kind regards,

Surinder K. Batra

Academic Editor

PLOS ONE

Additional Editor Comments (optional):

Reviewers' comments:

Reviewer's Responses to Questions

**Comments to the Author**

1. If the authors have adequately addressed your comments raised in a previous round of review and you feel that this manuscript is now acceptable for publication, you may indicate that here to bypass the “Comments to the Author” section, enter your conflict of interest statement in the “Confidential to Editor” section, and submit your "Accept" recommendation.

Reviewer #2: All comments have been addressed

2. Is the manuscript technically sound, and do the data support the conclusions?

Reviewer #2: Yes

3. Has the statistical analysis been performed appropriately and rigorously? 

Reviewer #2: Yes

4. Have the authors made all data underlying the findings in their manuscript fully available?

Reviewer #2: Yes

5. Is the manuscript presented in an intelligible fashion and written in standard English?

Reviewer #2: Yes

6. Review Comments to the Author

Reviewer #2: You have addressed all previous comments and I feel that the manuscript is now acceptable for publication

7. PLOS authors have the option to publish the peer review history of their article (what does this mean?). If published, this will include your full peer review and any attached files.

Reviewer #2: No

---

## [Editor Report · Acceptance letter]

3 Mar 2022

PONE-D-21-14241R2 

Upregulation of PODXL and ITGB1 in pancreatic cancer tissues preoperatively obtained by EUS-FNAB correlates with unfavorable prognosis of postoperative pancreatic cancer patients 

Dear Dr. Taniuchi:

I'm pleased to inform you that your manuscript has been deemed suitable for publication in PLOS ONE. Congratulations! Your manuscript is now with our production department. 

Kind regards, 

on behalf of

Prof. Surinder K. Batra 

Academic Editor

PLOS ONE